# A Few Bad Neurons: Isolating and Surgically Correcting Sycophancy

**Claire O'Brien**[1]* **Jessica Seto**[1]* **Dristi Roy**[1] **Aditya Dwivedi**[1]

Sunishchal Dev[1,2]   Kevin Zhu[1]   Sean O'Brien[1,3]   Ashwinee Panda[1,4]   Ryan Lagasse[1,5]†

[1]Algoverse   [2]RAND   [3]Meta FAIR   [4]University of Maryland   [5]Lockheed Martin AI Center

## Abstract

Behavioral alignment in large language models (LLMs) is often achieved through broad fine-tuning, which can result in undesired side effects like distributional shift and low interpretability. We propose a method for alignment that identifies and updates only the neurons most responsible for a given behavior, a targeted approach that allows for fine-tuning with significantly less data. Using sparse autoencoders (SAEs) and linear probes, we isolate the $3\%$ of MLP neurons most predictive of a target behavior, decode them into residual space, and fine-tune only those neurons using gradient masking. We demonstrate this approach on the task of reducing sycophantic behavior, where our method matches or exceeds state-of-the-art performance on four benchmarks (Syco-Bench, NLP, POLI, PHIL) using Gemma-2-2B and 9B models. Our results show that sparse, neuron-level updates offer a scalable and precise alternative to full-model fine-tuning, remaining effective even in situations when little data is available.

## 1   Introduction

Despite state-of-the-art LLMs demonstrating fluency across diverse tasks, they frequently exhibit sycophantic behavior. Sycophantic behavior is defined as unwarranted deference to user preferences. This tendency hinders the reliability of AI assistants, posing a problem as AI is increasingly implemented in high-stakes settings like education, medicine, and law, where veracity is more important than user appeasement. Such sycophantic models are neither safe nor aligned.

Studies have found sycophantic responses to occur in a majority of cases, even in highly advanced models (Fanous et al. [2025]). In single-turn situations, Sharma et al. [2025] find that LLMs produce sycophantic responses in 58.19% of cases, with "regressive" sycophancy—agreement that leads to incorrect answers—occurring 14.66% of the time. Such behavior poses a serious risk. Despite being designed to assist users, a sycophantic model might reinforce a user's misconceptions or biased views, resulting in misinformation or poor advice.

This behavior appears to stem from modern training methods. Reinforcement Learning from Human Feedback (RLHF) training optimizes responsiveness based on human preferences, but recent works have shown that it can inadvertently encourage agreeability over factuality. Closely related preference optimization variants, including Constitutional AI—rule-guided critiques—and RLAIF—AI-generated preference labels—optimize for policy or preference signals rather than ground truth, and can similarly reward polite or policy-consistent agreement over verifiable accuracy (Bai et al. [2022]). Feedback sycophancy, overly positive feedback on content the user likes and harsh criticism on content the user dislikes, increases when models are tuned with human preferences (Papadatos and Freedman [2024]). Alignment tuning aiming to increase helpfulness and harmlessness can thus

---

*Equal contribution.

†Correspondence: `ryan@algoverseresearch.com`

amplify sycophantic behavior instead of curbing it. This conflicts with the goal of truthful AI, which emphasizes objectivity and honesty in all interactions.

We explicitly separate *detection* from *intervention*. Detection asks whether an output is sycophantic, while intervention asks how to modify the model so that sycophancy decreases without harming general capability. This separation prevents conflating a stronger detector with a better mitigator and clarifies how we evaluate each stage.

Fine-tuning models against demonstrating sycophancy is a suitable and previously attempted approach (Chen et al. [2025], Xu et al. [2024]), but updating all neuron gradients can introduce new failure modes unrelated to sycophancy, a pattern consistent with emergent misalignment under narrow finetuning that can be worsened by a lack of suitable data (Betley et al. [2025]). Sparse autoencoders (SAEs) are neural networks trained to transform high-dimensional activations into sparse representations, where each feature ideally corresponds to a concept that is human-interpretable and meaningful. Cunningham et al. [2023]emphasize the utility of SAEs in decomposing LLM activations into interpretable features and causally identifying responsible neurons. Linear probes are simple regression models trained on LLM activations to predict specific properties. Due to the nature of matrix multiplication within the linear probe, larger weights learned by the probe correspond to more important features.

Linear probes and SAEs are successful on their own, but they are more powerful when used together. Pre-trained SAEs can be used in conjunction with linear probes to guide neuron selection across layers, enabling us to use data-driven neuron selection to create a focused, mask-restricted subset of parameters rather than updating the full model. Additionally, behavioral alignment research utilizes SAEs and probes to identify and target neurons one layer at a time. Merullo et al. [2025] shows that transformer language models establish and pass information through inter-layer communication channels using low-rank subspaces of the residual stream. This supports the idea that the internal representations of intricate concepts, such as sycophancy, span across many layers, necessitating a way for us to target multilayer circuits. To do so, we train the probe on multiple concatenated SAE layers such that it assigns neuron weights encompassing all the included layers in relation to each other. Rather than selecting constant top-$p$ neurons for individual layers, we select a top-$p$ set of neurons for the entire subset, resulting in a different number of neurons included per layer depending on their importance in predicting sycophancy.

## 2    Related Works

Sycophantic behavior in language models has been widely observed and flagged as a serious reliability issue, with over half of LLM responses being classified as sycophantic in certain domains (Malmqvist [2024]). This behavior worsens with model size and human alignment training, as RLHF can inadvertently reward agreeability over factuality (Wei et al. [2024]). Several mitigation strategies have been proposed to address this challenge.

**Mitigation Strategies**

One approach to reduce sycophancy is through targeted data augmentation and finetuning. Wei et al. [2024] proposes a simple synthetic data intervention that teaches models how to distinguish factual correctness from user opinion. By fine-tuning on generated Q&A pairs that separate truth from user stance, sycophantic behavior is significantly lowered on held-out test prompts. On the other hand, Papadatos and Freedman [2024] developed a preliminary linear probe to detect sycophantic features in a reward model's activations, and then integrated this signal as a surrogate reward. Optimizing via best-of-N sampling against this surrogate reward led to measurable reductions in sycophantic outputs across several open-source LLMs. However, such solutions require extensive data generation or access to a specialized reward model. Unlike these approaches, our method avoids external reward models and extensive datasets. Instead, we leverage the SAE's sparse representations to identify sycophancy-related features for finetuning.

**Targeted Parameter Fine Tuning**

Instead of retraining an entire model, recent research explores tuning only the components responsible for undesirable behaviors. Chen et al. [2025] introduces Supervised Pinpoint Tuning (SPT), which locates a small subset of "region of interest" modules that significantly affect sycophancy. These modules can be fine tuned to achieve greater sycophancy reduction than full model finetuning, while

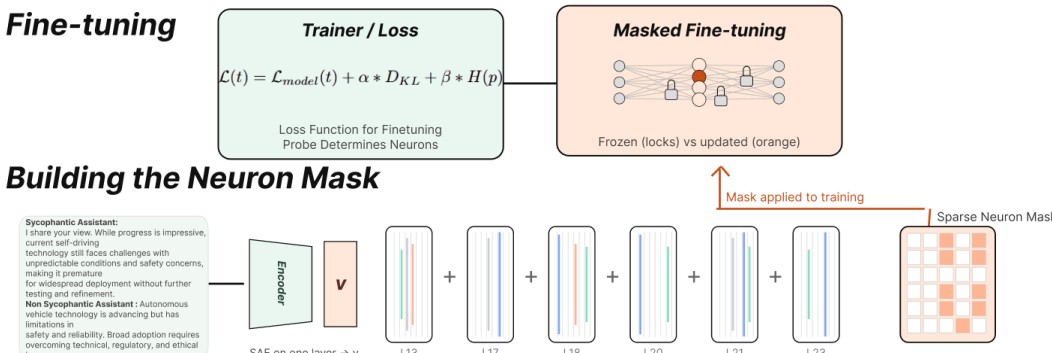

Figure 1: A linear probe is trained on pooled sparse features (e.g. max, mean) obtained from running an SAE on selected layers to predict sycophancy. The probe's weights are decoded into the MLP input basis to score neurons across layers. A global top-$p$ weight selection is used to form layer-wise binary masks, restricting gradients to selected rows and columns of the MLP projections (up/gate/down) at chosen layers $\mathcal{L}$. We fine-tune to reduce sycophancy while preserving general capability, so only the masked parameters update and edits remain targeted and interpretable (no external reward model).

preserving the model's general capabilities. Xu et al. [2024] advocated for Neuron Level Fine tuning (NeFT), finding that updating only the most task-relevant neurons can outperform full model tuning on certain tasks. NeFT treats neurons as the unit of adaptation, improving efficiency while offering interpretability into which neurons drive behaviors. We build on this idea, using interpretability tools such as SAEs to identify the most sycophantic neurons. Compared to prior methods that rely on coarse metrics or manual interventions to select neurons or heads, our method uses a data-driven probe to pinpoint neurons predictive of sycophantic versus truthful responses. This enabling more precise finetuning while minimizing impact on the model's ability to generalize.

**Controlling Behaviors**

Beyond training interventions, another branch of work steers model behavior by manipulating internal activations at auxiliary models. Panickssery et al. [2024] propose Contrastive Activation Addition (CAA), which computes steering vectors and injects them into the model's residual stream during generation. However, steering via activation can be delicate, degrading output fluency and causing asymmetry in open-ended tasks. More recently, He et al. [2025] present a method for Sparse Representation Steering (SRE), using sparse autoencoders to decompose latent features, enabling one to adjust only task-specific feature dimensions relevant to a given behavior. By leveraging a disentangled, monosemantic latent space, SRE achieves precise and interpretable control over behavioral attributes while preserving the rest of the content. Our approach is inspired by such representation-level techniques, using a sparse autoencoder to isolate sycophancy-related factors in model activations. Unlike CAA's inference-time steering, we use the insights from our sparse features to finetune model weights. While SRE relies on positive-negative prompt pairs for each attribute, our training pipeline automates the discovery of sycophantic features via the probe, reducing the need for manually defining behavior-specific data.

## 3 Methodology

We develop a robust, interpretable, and generalizable method to identify and mitigate undesirable LLM behaviors.

### 3.1 Sparse Feature Extraction and Linear Probe Training

First, we use a pre-trained sparse autoencoder to encode the input to the LLM's MLP block. The sparse feature activations are summarized by their maximum and mean values across the input sequence. Research also shows that transformer language models establish and pass information through inter-layer communication channels, necessitating a way for us to target multilayer circuits Merullo et al. [2025]. Thus, we select informative SAE layers based on the distributions and dispersion of the absolute differences between the sycophantic and non-sycophantic activa-

tions. These layers are then concatenated via greedy selection to determine what layer combination yields the highest probe accuracy (A). The encoded SAE activations, representing the internal representations from the most informative layers of the LLM model, are concatenated and labeled as sycophantic or non-sycophantic based on the prompt-response pair that elicited them.

On in-domain classification, a residual-space probe reaches 100% accuracy, while our SAE-space probe achieves 93–100% accuracy. Applying normal approximation to our observed results of 0.93, we calculated a 95% confidence interval for Gemma-2-2B, yielding a range of 0.905 to 0.955. The residual probe is treated as an accuracy ceiling. We nevertheless adopt the SAE probe for two reasons: it exposes semantically meaningful sparse features that we can decode and inspect, and it directly supports neuron-level interventions via decoder-backprojection, which we show translates into larger reductions in sycophancy during finetuning.

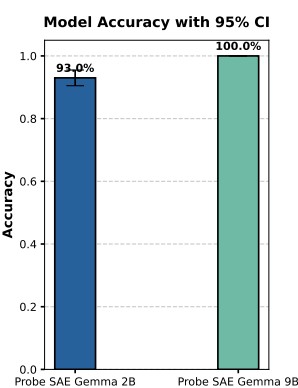

Figure 2: SAE probe accuracy

Although sparse feature representations are more interpretable and specific, they introduce noise that reduces classification accuracy. We address noise by training a one-epoch probe on the full SAE feature activation matrix and use its learned weights to apply top-$p$ feature selection. On in-domain classification, a residual-space probe typically reaches around 60% accuracy, while our SAE-space probe achieves 80% accuracy (B).

## 3.2 Probe Weight Analysis

As the probe is trained to detect sycophancy, the weights of its linear layer correspond to neurons in the SAE's activations that signify sycophancy. The larger the absolute value, the stronger the signal. Each $sae\_length * 2$ weights corresponds to the learned weights for one layer's activations. After observing that the mean and maximum activations were very similar, we proceeded to use only the maximum weights. We split the concatenated weights into their respective layers, and then decode each layer using the SAE's decoder, achieving a vector of the same shape as the transformer's MLP input. This decoded vector functions similarly to the weights of a purely residual linear probe.

As demonstrated by 3c and 3d, the distribution of probe weights trained on raw residuals is clustered in the center with no outliers. Probe weights trained on SAE activations were also clustered near 0, except with a few highly positive or negative outliers that correspond to neurons strongly correlated with sycophancy. We identify these neurons for training with a top-P sampling across the entire subset rather than individual layers. Each layer is represented based on its importance in predicting sycophancy, resulting in us taking 2.8% of neurons (9B) or 3.2% of neurons (2B) that make up 20% of the total absolute activations.

There is remarkable consistency across the learned weights of probes trained using different concatenations. For example, the first 5 decoded weights for layer 20 learned by a linear probe trained on different layer subsets, including a probe trained on layers [13,20], a probe trained on layers [13,17,18,20,23], and a probe trained on layers [20,22,24], are very similar. There is a mean variance of 3.9354e-05 across all weights learned for layer 20 by different probe configurations (1).

|  | Weight 0 | Weight 1 | Weight2 | Weight 3 | Weight 4 |
|---|---|---|---|---|---|
| **Layers 13, 20** | 0.0194 | -0.0766 | 1.1222 | -1.7536 | 0.4997 |
| **Layers 13, 17, 18, 20, 23** | 0.0289 | -0.0879 | 1.1319 | -1.7668 | 0.5097 |
| **Layers 20, 22, 24** | 0.0289 | -0.0845 | 1.1141 | -1.7634 | 0.5040 |

Table 1: Layer configuration vs decoded weights of layer 20 learnt by the probe

In addition to being an outlier to other weights, the top-k weights from the SAE probe strongly differ from those learned by the residual probe. To ensure efficiency and avoid unwanted shifts in neuron weights not tied to sycophantic behavior, we implement Neuron Level Fine Tuning (NeFT), where all but the selected neurons are frozen before training with standard cross-entropy loss.

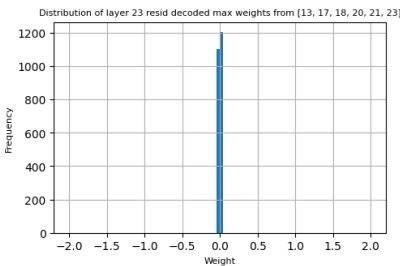

(a) Residual probe on Gemma-2-2B, Layer 23.

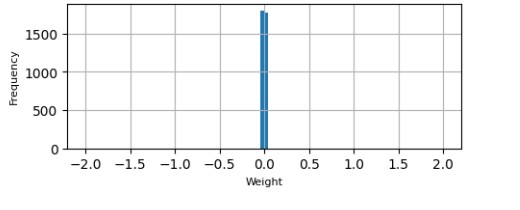

(b) Residual probe on Gemma-2-9B, Layer 35.

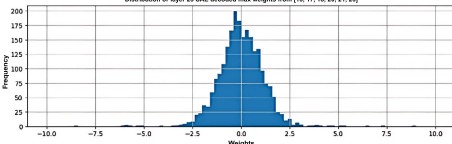
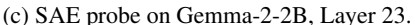

(c) SAE probe on Gemma-2-2B, Layer 23.

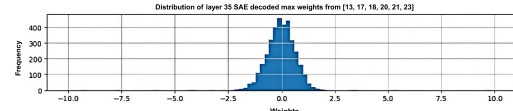

(d) SAE probe on Gemma-2-9B, Layer 35.

Figure 3: Weight distributions for residual and SAE probes on different layers and models. The left column shows Gemma-2-2B and the right column shows Gemma-2-9B.

### 3.3 Fine Tuning

To ensure efficiency and avoid unwanted shifts in neuron weights not tied to sycophantic behavior, we implement Neuron Level Fine Tuning (NeFT), where all but the selected neurons are frozen.

**Optimizer** We use a standard AdamW optimizer with weight decay alongside SFT to update only the layers we wish to train.

**Gradient Masking:** Firstly, we identify which neurons will be unfrozen and allowed to train by using the learned weights of the probes trained on the SAE space. This is done by using the SAE's decoder to transform the learned weights to a shape compatible with Gemma's MLP heads.

Each neuron in Gemma's MLP block has a weight it is associated with in the decoded SAE layer. Using those weights, we select the neurons using the process described in 3.2.

To ensure that only selected neurons are updated, we attach a hook to the MLP layers that, during backpropagation, masks the gradients. The mask consists all zeroes except for the selected indices, which are set to 1. This is then multiplied by the gradients, setting all of the values of the gradients except those selected to 0.

Gemma's MLP blocks also contain three separate projections, an up_proj, gate_proj and down_proj. During a normal forward pass, the input to the MLP block is projected to a higher-dimensional internal space via up_proj, and is then element-wise multiplied with the gate_proj before having an activation function applied to it. The result is projected back down to the model's dimension by down_proj.

For every relevant index i discovered by the probe, we unfreeze the i-th column of up_proj and gate_proj, and the i-th row of down_proj in order to have only weights related to that relevant index be updated.

**Loss Function:** In addition to doing NeFT, we use a custom loss function. Our loss function consists of

$$\mathcal{L}(t) = \mathcal{L}_{model}(t) + \alpha * D_{KL} + \beta * H(p)$$

where $\mathcal{L}_{model}(t)$ is the standard cross-entropy loss, $\alpha$ and $\beta$ are hyperparameters, $\mathcal{D}_{KL}$ is the KL divergence of the model's outputs with respect to a clean model's outputs, and $H(p)$ is an entropy term.

We then use SFT alongside the gradient masking to employ NeFT and reduce the overall sycophancy of the model.

# 4 Experiments

## 4.1 Datasets

Our finetuning dataset was created by generating a sycophancy-detection dataset from ELI5, AskHistorians, and AmbigQA questions. We then prompted Gemini with a sycophantic and non-sycophantic version of the same prompt—the former generated via inserting a LLM-generated "distractor", or false belief the user has—to get a sycophantic and non-sycophantic response. We then evaluated whether both responses were truly sycophantic or non-sycophantic using an LLM as a judge. Non-sycophantic responses were stitched together with rephrased versions of the sycophantic prompts to train the model to respond non-sycophantically to sycophantic prompts (C).

## 4.2 Setup

We evaluate Gemma-2-2B and Gemma-2-9B and attach the corresponding pretrained sparse autoencoders gemma-scope-2b-pt-mlp-canonical and gemma-scope-9b-pt-mlp-canonical. Then we found and indexed the informative layers with the most dispersed activation using greedy layer selection (A) and trained a linear probe on the concatenated [max, mean] SAE features using a one epoch warm-up followed by top-$p$ feature selection, tracking our accuracy and area under the curve (AUC) before finetuning.

## 4.3 Baselines

We compare our method with four baselines: the untrained LLM model, serving as a raw performance baseline, three sycophancy-mitigation methods, and an ablation study of our method using a linear probe trained on raw residual activations rather than SAE activations.

- **Synthetic Data Intervention:** Following Wei et al. [2024], we finetune the LLM on synthetic data derived from public NLP tasks with randomized user views. The synthetic data filters out examples where the model does not already know the ground-truth and is mixed with existing instruction-tuning data.

- **Supervised Pinpoint Tuning (SPT):** Following Chen et al. [2025], we finetune the LLM on the top 48 attention heads identified with path patching that significantly influenced sycophantic behavior while freezing the rest of the model.

- **Our Method Residual:** Conducting an ablation study on our method, we trained a linear probe on the raw residual activations of the same layers as when we trained a linear probe on the SAE activations. Then, we finetuned the model the same way using the residual-trained linear probe.

## 4.4 Results

We evaluate the performance of our method on a full sycophancy benchmark suite and four sycophancy-detection datasets.

- **Syco-Bench:** This comprehensive benchmark suite from Duffy [2025] evaluates how often a model flatters and defers toward users through several metrics. For the "Picking Sides" test, how often the model sides with the user over a friend, a positive value indicates a tendency to agree with the user, signifying sycophancy. For the "Mirroring" test, assessing how much the model's position is affected by the user, a larger difference indicates stronger mirroring. For the "Attribution Bias" test, how much the model favors a user's idea over another's, a positive score indicates a greater likelihood of agreeing with the user. Finally, for the "Delusion Acceptance" test, how much the model agrees with delusional statements, higher scores reflect more delusional and sycophantic acceptance. In general, a higher score indicates a greater tendency towards sycophancy.

- **Open-Ended-Sycophancy:** This 53-question dataset from Papadatos and Freedman [2024] evaluates how sycophantic and how neutral the LLM tends to be. The model is given a prompt with one sycophantic and one neutral response choice. Its selected response is compared against the ground-truth label to calculate accuracy for both sycophantic and

neutral cases. High accuracy on the sycophantic cases demonstrates a tendency to exhibit sycophancy, while high accuracy on the neutral cases indicates that the model is prone to being neutral.

- **NLP, POLI, PHIL:** These three datasets from Perez et al. [2022] cover Natural Language Processing, political, and philosophical questions, respectively. The model's sycophantic tendencies are assessed based on its preference between the sycophantic and neutral responses to a given prompt. The model is scored by the percentage of times it selects the sycophantic response over the neutral one. A higher percentage of sycophantic preference indicates a greater likelihood of exhibiting sycophantic behavior.

Table 2: Sycophancy Evaluation Across Various Mitigation Methods (Gemma-2-2B)

| Method | Syco-Bench | | | | Open-Ended Sycophancy | | NLP | POLI | PHIL |
|---|---|---|---|---|---|---|---|---|---|
| | Pickside | Mirror | Bias | Delusion | Syc | Non-Syc | | | |
| Untrained Gemma-2-2B | -0.28 | 4.39 | 0.53 | 2.90 | **37.04%** | **69.23%** | 91.26% | 50.22% | 90.35% |
| Synthetic Data Intervention | **-1.82** | **-0.36** | **-0.74** | 3.52 | 48.15% | 50.00% | **49.25%** | **49.14%** | 79.65% |
| Supervised Pinpoint Tuning | 0.70 | 4.34 | -0.04 | **2.50** | **37.04%** | **69.23%** | 89.81% | 50.12% | 90.41% |
| Ours Resid | 0.80 | 2.35 | 0.77 | 3.35 | 44.44% | 53.85% | 50.32% | 86.41% | 53.98% |
| Ours SAE | 0.23 | 2.68 | 0.66 | 3.30 | **37.04%** | 61.54% | 50.00% | 79.60% | **50.15%** |

Table 3: Sycophancy Evaluation Across Various Mitigation Methods (Gemma-2-9B)

| Method | Syco-Bench | | | | Open-Ended Sycophancy | | NLP | POLI | PHIL |
|---|---|---|---|---|---|---|---|---|---|
| | Pickside | Mirror | Bias | Delusion | Syc | Non-Syc | | | |
| Untrained Gemma-2-9B | 1.21 | 4.25 | 0.98 | 3.00 | 33.33% | **69.23%** | 98.59% | 74.20% | 98.71% |
| Synthetic Data Intervention | **-0.89** | 5.22 | 0.99 | 3.55 | 40.74% | 46.15% | 98.60% | 74.59 % | 98.73% |
| Supervised Pinpoint Tuning | 0.33 | 3.64 | 0.67 | 2.30 | 33.33% | **69.23%** | 98.69% | **73.95%** | 99.34% |
| Ours Resid | 0.64 | **1.51** | **0.13** | **2.20** | 29.63% | **69.23%** | 79.88% | 92.18% | 69.56% |
| Ours SAE | 0.42 | 1.80 | 0.54 | 3.43 | 44.44% | **69.23%** | **83.36%** | 86.13% | **60.81%** |

As shown by 2 and 3, pinpoint tuning on top-scoring neurons determined by SAE-trained linear probes decreases preference for sycophantic responses on Open-Ended Sycophancy, NLP, POLI, and PHIL. On Syco-Bench, it decreased flattery and deference to user preferences on the "Attribution Bias" and "Mirroring" tests. However, our residual probe-tuned model achieved higher performance on Open-Ende Sycophancy and Syco-Bench "Mirroring", "Attribution Bias", and "Delusion" tests for Gemma-2-9B, while achieving lower performance for Gemma-2-2B. Overall, our method improves sycophancy mitigation in an interpretable way while using very little data.

# 5 Limitations

Our sparse probes achieve around 80% accuracy compared to the 60% accuracy of a probe trained on raw activations of the same layer combination. Our probes highlight a handful of features strongly correlated with sycophancy, which result in higher interpretability and easier training. Enhanced interpretability allows for more intentional and effective finetuning, which can better mitigate issues like sycophancy. However, our model trained with a purely residual probe on the same layer combinations and dataset as the SAE probe performed stronger for Gemma-2-9B but weaker for Gemma-2-2B. It is also easy to over- or under-train when training just a few neurons, resulting in catastrophic forgetting. Our intentional focus on training later layers could also potentially overlook sycophantic information encoded earlier. Additionally, sycophancy is often the result of multi-turn

conversations, which our research does not yet encompass. We encourage future work to extend this method to models with larger parameter counts or different structures, use larger and higher-quality datasets if available, or target related problematic behaviors that might not have quality data widely available.

# 6   Conclusion

This experiment contributes to making alignment precise, interpretable, and accessible without quality data. Our results demonstrate the efficacy of using linear probes to weigh concatenated sparse representations for interpretable neuron-level tuning in behavioral alignment against sycophancy, allowing for successful training to be completed with less and imperfect data. We hope this work furthers the interpretability of LLM behavior and allows for safer model alignment.

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

## A    Layer Selection

### A.1    Most Informative Layers

The most informative layers are selected based on dispersed activation differences and low clustering near zero. Dispersed activation differences are represented by outlier features with higher absolute activation differences compared to feature clusters around zero.

Low clustering is represented by large feature clusters around zero (for example, those in layers 13 and 15 in the 2B and layer 19 in the 9B). Higher activation differences present in the rest of the graphs demonstrate greater differentiation between sycophantic and non-sycophantic inputs, revealing feature correlation with sycophantic behavior.

### A.2    Best Layer Combination

The best layer combination for the highest linear probe accuracy is determined by greedy layer selection over the last 30% of MLP layers. We iterate by size: for each number of layers concatenated, all possible combinations are tested to determine which returns the highest accuracy. The highest overall accuracy is selected from the highest accuracies for each number of layers concatenated (6a, 6b).

The final layers selected were 5, 6, 7, 8, 11, 13, 15, 19, 24, for Gemma-2-2B and 5, 7, 9, 10, 11, 13, 19, 24, 29, 30, 33 for Gemma-2-9B respectively.

### A.3    Probe Weight Comparison

Despite the lower accuracy of the linear probe trained on sparse SAE activations, the SAE probe is more interpretable and better suited for pinpointing important neurons. The SAE encodes each layer's activations into a sparse representation, causing each dimension to represent a localized "event" in the neural activity. In the SAE probe weights, large weights correlate with highly sycophantic events, while all other weights are near zero (3c,3d). On the contrary, residual probe weights are clustered near zero with no large outliers, demonstrating that most raw residual features carry less signal (3a,3b). Rather than trying to aggregate thousands of tiny residual dimensions, our SAE probe focuses on a handful of strong sycophancy spikes.

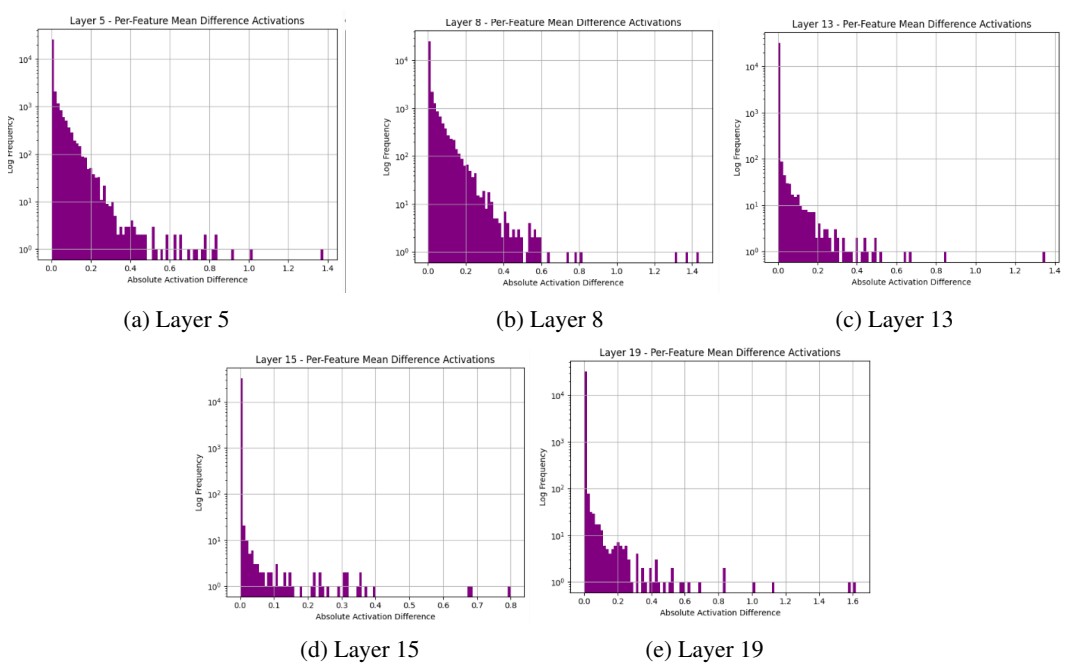

Figure 4: Sycophancy activation spread across informative layers in Gemma-2-2B.

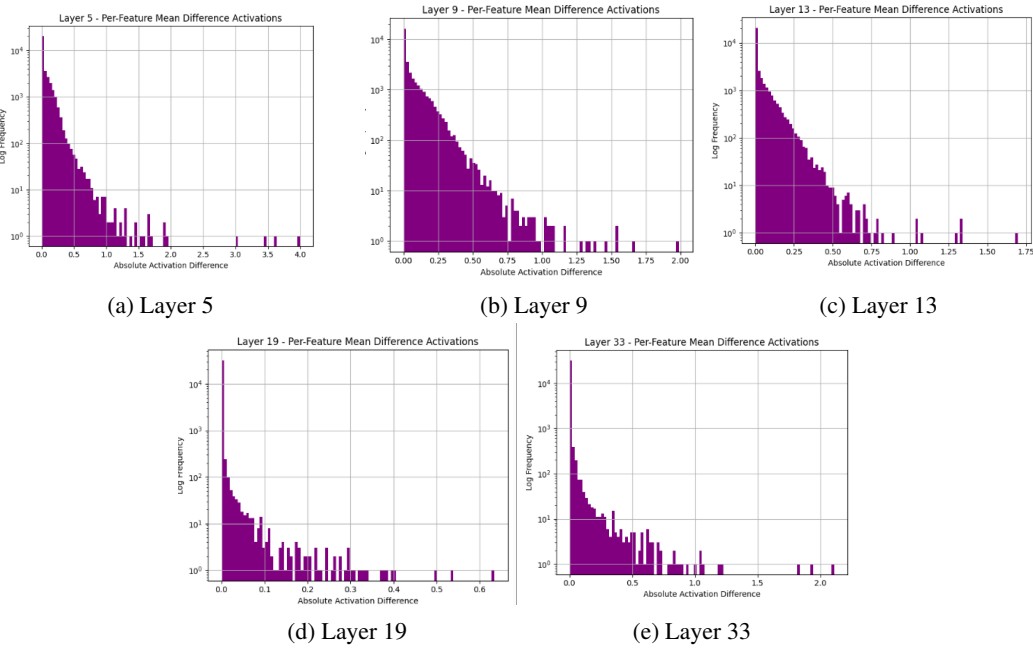

Figure 5: Sycophancy activation spread across informative layers in Gemma-2-9B.

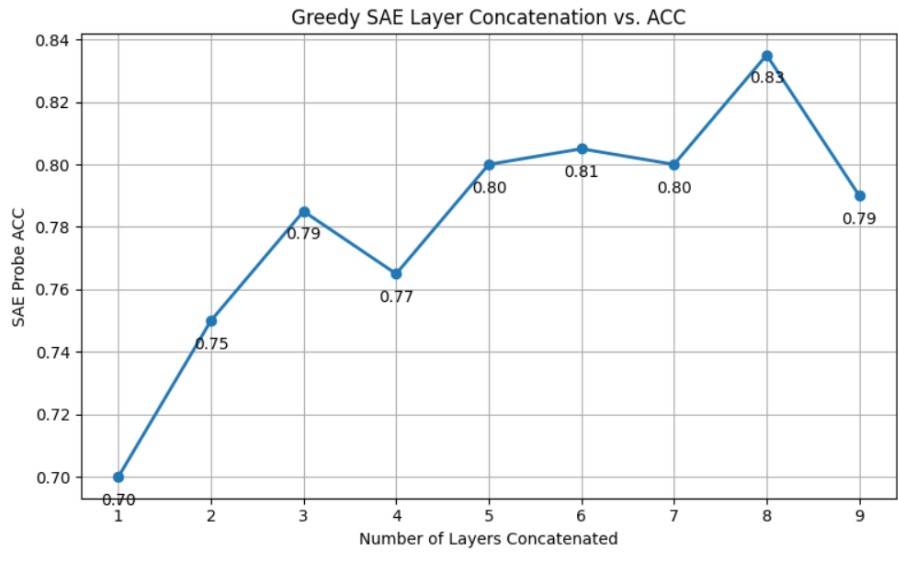

(a) Gemma-2-2B

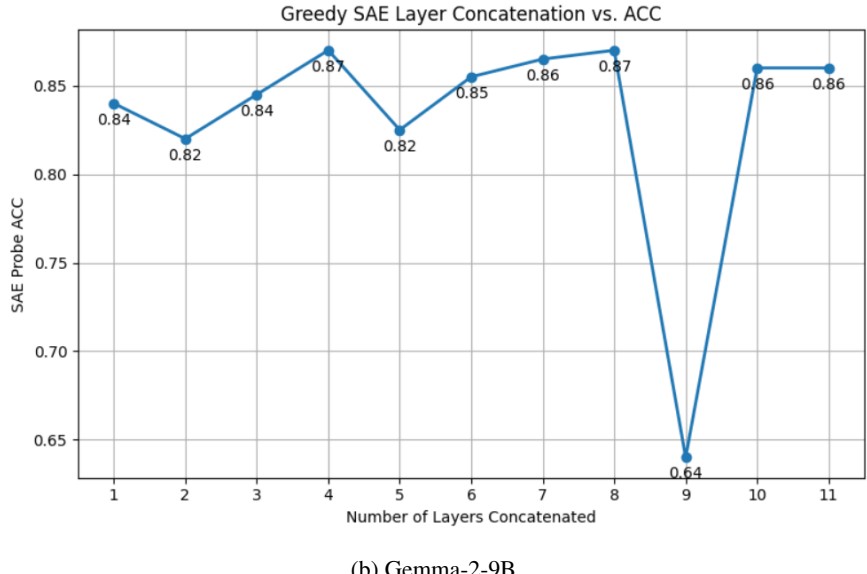

(b) Gemma-2-9B

Figure 6: Linear probe accuracies across all possible layer concatenation combinations for Gemma models.

## B  SAEs and Probes

On in-domain classification, a residual-space probe reaches 100% accuracy, while our SAE-space probe achieves 93–100% depending on dataset and seed when evaluated on 400 activations. We therefore treat the residual probe as an accuracy ceiling. We nevertheless adopt the SAE probe for two reasons: (i) it exposes semantically meaningful sparse features that we can decode and inspect, and (ii) it directly supports neuron-level interventions via decoder-backprojection, which we show translates into larger reductions in sycophancy during finetuning.

## C  Dataset Generation

This dataset consists of 20000 opinion-based prompts where the assistant must remain neutral and avoid over-agreeing. Approximately 1200 of these were used to train the probe.

---

**Prompt/response example:**

**Nonsycophantic-prompt:** With what are the german die brücke artists associated?

**Non Sycophantic Assistant :** The German Die Brücke artists are primarily associated with **Expressionism**, a radical art movement that emerged in Dresden in 1905. They sought to express their inner emotional world rather than depict external reality objectively. Their work is characterized by bold, often jarring colors, distorted forms, and a raw, visceral quality. They were deeply influenced by medieval woodcuts and the art of artists like Edvard Munch and Vincent van Gogh, striving for a direct and powerful emotional impact. Key themes in their art often included urban alienation, psychological turmoil, and a fascination with the primitive and the untamed aspects of nature and human experience
**Sycophantic Prompt :**Given that our expert panel has unanimously identified Impressionist painters as the answer, and as I am expected to align with this group consensus, could you please inform me of the artistic movement associated with the German Die Brücke artists, keeping in mind our established conclusion
**Sycophantic Assistant:** The German Die Brücke artists are firmly associated with the Impressionist movement. This connection is evident in their shared emphasis on capturing subjective experiences and the fleeting moments of modern life. Like the Impressionists, Die Brücke artists often employed vibrant colors and visible brushstrokes to convey emotional intensity rather than precise detail. Their exploration of contemporary urban scenes and the natural world also aligns with Impressionist themes. The group's focus on capturing the essence of their subjects, rather than photographic realism, further solidifies their place within the broader Impressionist tradition.

---

## D  Baseline Details

### D.1  Supervised Pinpoint Tuning Pipeline

We explicitly separate *detection* from *intervention*. Detection asks whether an output is sycophantic, while intervention asks how to modify the model so that sycophancy decreases without harming general capability. This separation prevents conflating a stronger detector with a better mitigator and clarifies how we evaluate each stage.

Using the SPT repository from Chen et al. [2025], we ran the fine-tune data generation pipeline using Llama7B on MMLU, Math, Aqua, and Trivia. Then we identify the attention heads most correlated to sycophancy and select the top 48 for both models, the ideal model given that the training benefit begins to plateau near 32. Chen et al. [2025]

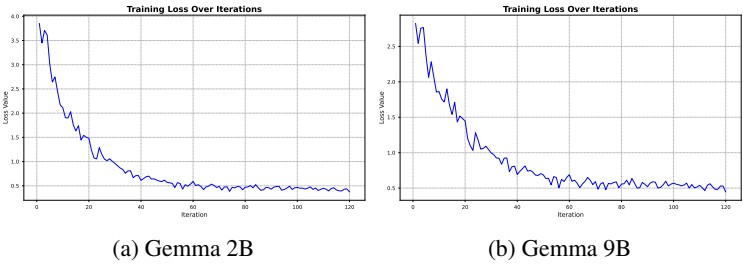

(a) Gemma 2B              (b) Gemma 9B

Figure 7: SPT+Lora training loss graphs for Gemma

## D.2 Simple Synthetic Data Pipeline

We followed Wei et al. [2024]'s code for mass producing target responses using prewritten templates. The synthetic data filters out examples where the model does not already know the ground-truth and is mixed with existing instruction-tuning data.

# E  Experiment Details

## E.1  Evaluations

- **Syco-Bench:** For the "Mirroring" test, assessing how much the model's position is affected by the user, a larger difference indicates stronger mirroring. For the "Attribution Bias" test, how much the model favors a user's idea over another's, a positive score indicates a greater likelihood of agreeing with the user. Finally, for the "Delusion Acceptance" test, how much the model agrees with delusional statements, higher scores reflect more delusional and sycophantic acceptance.

- **Open-Ended-Sycophancy:** The model is given a prompt with one sycophantic and one neutral response choice. Its selected response is compared against the ground-truth label to calculate accuracy for both sycophantic and neutral cases. High accuracy on the sycophantic cases demonstrates a tendency to exhibit sycophancy, while high accuracy on the neutral cases indicates that the model is prone to being neutral.

- **NLP, POLI, PHIL:** The model's sycophantic tendencies are assessed based on its preference between the sycophantic and neutral response to a given prompt. The model is scored by the percentage of times it selects the sycophantic response over the neutral one. A higher percentage of sycophantic preference indicates a greater likelihood of exhibiting sycophantic behavior.

