# OpenReview forum: "A Few Bad Neurons: Isolating and Surgically Correcting Sycophancy"
_NeurIPS.cc/2025/Workshop/Reliable_ML — NeurIPS 2025 - Reliable ML Workshop_

### Official Review · Reviewer_u7Ao · 2025-09-19
**Promising efficient LLM alignment method while stronger evaluation is needed**

**Rating:** 7
**Confidence:** 3

**Review:**

## Summary

This paper proposes an LLM alignment method that combines sparse autoencoder (SAE) with neuron-level fine-tuning (NeFT) to identify, isolate, and update the parameters most related to unwanted behavior. Specifically, this paper studies sycophantic behavior, which means LLMs defer to user preference and sacrifice factual correctness.

Using sparse autoencoder (SAE), the behavior of LLMs is encoded into sparse and semantically separated activations. Linear probes are trained on top of these activations to identify neurons related to sycophancy. Then, these neurons will be fine-tuned to alleviate sycophancy while other neurons are frozen for stability.

Experiments on Gemma-2-2B and 9B across multiple benchmarks (Syco-Bench, Open-Ended-Sycophancy, NLP, POLI, and PHIL) show that the proposed method matches or outperforms synthetic data intervention and supervised pinpoint tuning, while requiring significantly less data. However, the evaluation is not overwhelmingly strong, with the proposed method underperforming in some aspects.

## Strengths

1. **Promising idea for interpretable alignment**: The methodology provides a promising way to not only mitigate sycophancy but also provide insight into the responsible neurons for such behavior. The proposed method is also data-efficient and parameter-efficient.
2. **Extensive baselines and benchmarks**: The evaluation includes timely and diverse baselines and benchmarks, reflecting the latest advancements in the domain.
3. **Relevant to workshop topic**: The proposed method targets sycophancy mitigation for LLMs with access to small and partially synthetic datasets.

## Weaknesses

1. **Relatively weak empirical support and simple analysis**. The empirical results on Gemma-2-2B do not seem very strong, with the proposed method not performing better than the untrained model on Pickside, Delusion, and Open-Ended Sycophancy. For Gemma-2-9B, it is also worse than the untrained model on Pickside and Mirror. The advantage of the proposed method is not very convincing. The analysis is brief and confusing; for example, L266-268 should be about the proposed method instead of pinpoint tuning, and how the numbers are calculated is unclear to me.
2. **Incremental novelty**. Methodologically, the paper is largely an integration of existing methods (SAE and NeFT). The contribution lies more in application and empirical validation rather than theoretical and conceptual novelties.
3. **Lack of formulation and definitions**. Although the paper has cited the adopted methods, SAE and NeFT, the formulation of the methodology is too few to grasp an idea of how the proposed method works.

## Suggestions for Authors

1. **Add experiments for stronger claims**. While this paper claims to require fewer data and update fewer parameters, the experiments do not strongly demonstrate it. Consider sweeping over the size of data and percentage of parameters updated to strengthen the claim and ablate the sensitivity.
2. **Add more ablations**. The influence on generation quality is not measured (in terms of perplexity, downstream accuracy, etc.). Also, the sensitivity to design choices (such as layer selection) can be ablated.
3. **Expand the scope of undesired behaviors**. It seems to me that the proposed method is not restricted to sycophancy but can also be applied to hallucination, bias, etc. Adding experiments on these behaviors would add to the strength of the paper. Even if the proposed method does not perform well on certain types of behaviors, if insights about these failure modes can be provided, it will also be valuable.
4. **Improve the writing and formatting**. First of all, clearly indicate the type of elements referred to (Figure, Table, etc.). Current writing can be confusing. Also, explain the notations in Tables. In Table 2 and 3, the meaning of bold and underlined numbers is never explained. It would be nice to use arrows to indicate the desired direction of metrics when you have many.

---

### Official Review · Reviewer_5WVU · 2025-09-24
**Review response**

**Rating:** 6
**Confidence:** 2

**Review:**

The paper proposes a clear, surgical alignment approach: use SAEs + a linear probe to pick “sycophancy-predictive” MLP neurons and fine-tune only those via gradient masking; the setup is well-motivated and data-efficient for imperfect-label regimes.

## Weakness / Suggestions
- Large portions of probe and finetuning data are model-generated (GPT-4o) and probe-labeled; this risks target leakage and confirmation bias. Human ground truth and inter-annotator agreement are missing.
- The direction of the metric (Table 2, 3) is a bit confusing.  Would be helpful to state, for every metric, which direction is “less sycophancy,” and include deltas ±CI.